# The Coupling and Coordinated Development from Urban Land Using Benefits and Urbanization Level: Case Study from Fujian Province (China)

**DOI:** 10.3390/ijerph17165647

**Published:** 2020-08-05

**Authors:** Kun Wang, Yingkai Tang, Yaozhi Chen, Longwen Shang, Xuanming Ji, Mengchao Yao, Ping Wang

**Affiliations:** 1Business School, Sichuan University, Chengdu 610064, China; liam_wang@stu.scu.edu.cn (K.W.); tang@scu.edu.cn (Y.T.); chenyz@stu.scu.edu.cn (Y.C.); 2Mathematics School, Sichuan University, Chengdu 610064, China; 2017141211112@stu.scu.edu.cn; 3School of Finance and economics, Jimei University, Xiamen 361021, China; jxm@jmu.edu.cn; 4Economic School, Northwest Minzu University, Lanzhou 730030, China; mcyao@xbmu.edu.cn

**Keywords:** land use benefits, urbanization, Gini coefficient weighting method, coupling coordination degree (CCD), Fujian Province

## Abstract

In recent years, urbanization has been developing rapidly. However, it is also accompanied by land management problems, such as low land use efficiency. In this research, we manage to explore the temporal and spatial evolution laws as well as characteristics of the coupling and coordinated development between urbanization and land use benefits. Through this, it is possible for us to provide policy recommendations for the sustainable development of the urbanization in Fujian Province. In this study, we take prefecture-level municipal districts and county-level cities in Fujian as the research subject. We construct an index system, based on data in 2002, 2005, 2010, 2015, and 2017, to evaluate the urban land use benefits and urbanization. Besides, we leverage the Gini coefficient weighting method to give weight to each index and calculate the value of its benefits. Moreover, it is the relative development degree and the coupling coordination degree model that we comprehensively leverage to study the spatiotemporal evolution law of the coupling coordination degree (CCD). The results show that: (1) Urban land use benefits and urbanization level are positively correlated with the regional administrative level and economic development status; (2) The CCD of urban land use benefits and urbanization level in various regions of Fujian is still low. However, the overall development direction is good; (3) From the perspective of spatial distribution, the CCD owns a “center-periphery” pattern that is based on the law of diminishing CCD power from three central cities of Fuzhou, Xiamen, and Sanming. Consequently, it requires governments to take action. Firstly, they should promote the intensive land use in the urbanization process. Meanwhile, they should also pay attention to ecological environment protection. Besides, it is recommendable to give full play to the radiating and leading effect of central cities on surrounding ones. Finally, they are required to provide appropriate policies and resource support to peripheral cities.

## 1. Introduction

As a basic carrier of human activities and urban operations, land resources are a basic guarantee of urban socioeconomic stability. Their effects on urbanization have been gradually strengthening. The land resources are not only closely related to the development of urbanization itself, but also to the food security and social stability [1,2,3]. In recent years, the rapid progress of China’s urbanization has brought huge economic benefits. Nevertheless, it has developed at the expense of the ecological environment. More specifically, it has brought various environmental problems, such as ecological deterioration, resource scarcity, land degradation, overall climate change, and a comprehensive decline in urban land use. Even in some rural areas, there has been a phenomenon of abandoned farmland [4,5,6,7,8,9]. As a result, these problems have heavily affected the sustainable development of the regional economy and society as well as human well-being. What is worse, this accelerated urbanization process demands a lot of land resources [10]. Urban expansion in the peripheral areas of major city centers generally takes place at the price of prime farmlands [11]. China’s land for construction is still increasing with rapid urbanization [12]. Consequently, how to strike a balance among land uses, expand the benefits of land use while achieving healthy development is a hard problem lying in the process of China’s urbanization.

The land use benefits refer to economic, social, ecological, and environmental benefits. These kinds of benefits are directly generated by the utilization of the unit land area in a certain period and area [13]. They are the comprehensive benefits of all four benefits [14]. Nowadays, research on land use benefits is no longer only confined to the evaluation of land use benefits. However, they also include the relation between land use benefits and other factors. Foreign scholars’ research on urbanization and land use benefits is mostly independent. The majority regard cities as the research background on land use benefits, or as a way to optimize the urban environment. They mainly studied the impact of changes in the land use structure on cities under urbanization. They also focused on the interrelation between urban land use and urban growth [15,16,17,18,19,20,21,22]. Some also analyzed changes and benefits of land use by methods such as Geographic Information System (GIS), remote sensing technology, and space measurement [23,24,25]. Comparatively, in China, there are many studies on correlation. These studies mainly focused on the coupling relation between intensive land use and urbanization level [26,27,28,29,30], coordinated relation between intensive urban land use and socioeconomic development [31,32,33], as well as coupling and coordination relation between urbanization and urban land use benefits, etc.

Coupling is a concept in physics, referring to the motion of two (or more than two) systems affecting each other through interplay [34]. we often leverage it to measure the extent of the interplay among systems or motion. Comparatively, the coordination degree refers to a measure of the coordination degree among systems [35]. It can reflect whether the systems promote each other at a high level or restrict each other at a low one. Liao et al. first combined the degree of coupling with that of coordination. He proposed a model to measure the coupling and coordination degree (CCD) in a system or among multiple ones. What he has done made up for the defect that the coupling degree can only reflect the degree of interaction among systems, but not the level of development [35]. From then on, the CCD model has been widely used to evaluate the relationship between urbanization and land use benefits. Some scholars performed a time series analysis to study the coupling and coordination relation between urbanization and urban land use benefits. In their research, they took the major cities in China, such as Wuhan, Xining, Jinan, Shenzhen, as the research object [36]. However, most conducted research based on specific areas, such as the economic belt of Bohai Rim [37], Beijing–Tianjin–Hebei [38,39], Shaanxi–Gansu [40], three provinces in the Northeast [3], Middle Yangtze River Region [41], Guangxi Northern Gulf Economic Zone [42], as well as provinces of Shanxi [43], Zhejiang [44], Shenzhen [45], Jinan [46], etc. The main research methods include: entropy method [40,46], coefficient of variation method [3], gray correlation model [47], analytic hierarchy process (AHP) [45,48], data envelopment analysis (DEA) [49], Technique for Order Preference by Similarity to an Ideal Solution (TOPSIS) model [43] global principal component analysis [29,32], etc. It is found that the land use benefits and urbanization in many cities are still uncontrolled coordination. It is of significance to study the coupling and coordination relationship between the two. That is because it can, to some extent, prevent the urban area from developing in the form of “BIG PIE”. Consequently, it can also help realize the maximum comprehensive benefits of urban society, economy, ecology, and environment management. From the above analysis, we can conclude that in terms of research areas, researchers mainly focused on the more developed regions on the southeast coast, northeast as well as central and western regions. There are fewer studies on the relatively less advanced regions along the southeast coast. Fujian is the core area of the 21st Century Maritime Silk Road supported by the Party Central Committee [50]. Excellent geographical location and policy support in Fujian promote its development greatly. Nevertheless, its unique geographical feature of “eight mountains, one river, and one field” demonstrates the lack of available land resources. This defect severely limits the socioeconomic development and urbanization process of Fujian. Consequently, the intensive utilization of land, which contributes to the effective coordinated development of land use benefits and urbanization, is particularly significant for promoting the socioeconomic advancement of Fujian.

In view of the above theoretical and practical background, this study, taking prefecture-level municipal districts and county-level cities in Fujian Province as the research object, builds an index system. This system is based on statistical data in 2002, 2005, 2010, 2015, and 2017, to evaluate the urban land use benefits and urbanization level. We also comprehensively leverage the Gini coefficient weighting method and the CCD model to study the spatial and temporal evolution of the coupling coordination degree (CCD). The contributions of this paper are as follows. Firstly, the utilization of the new weight determination method, the Gini coefficient weighting method (GCWM), can reflect the information more fully. Secondly, this paper, taking county-level cities and prefecture-level municipal districts as the research object, reflects the local situation more accurately.

## 2. Data and Methodology

### 2.1. Study Area

Fujian Province is situated at 23°33′ N–28°20′ N and 115°50′ E–120°40′ E southeastern China. It is on the coast of the East China Sea, across the Taiwan Strait, and opposite to Taiwan. It is an important estuary in mainland China (see Figure 1), adjacent to Zhejiang Province in the northeast, crossing the Wuyi Mountains and bordering with Jiangxi Province in the northwest, connecting with Guangdong Province in the southwest, as well as connecting the Yangtze River Delta and the Pearl River Delta. As of the end of 2017, Fujian Province has jurisdiction over 9 prefecture-level cities of Fuzhou, Xiamen, Quanzhou, Zhangzhou, Putian, Longyan, Sanming, Nanping, and Ningde (for convenience, we all call the districts of prefecture-level cities below according to the names of prefecture-level cities. For example, we call Fuzhou city as Fuzhou). There are a total of 29 districts. Fujian spans about 124,000 square kilometers of land, with 14 county-level cities and 44 counties. It has a permanent population of 39.11 million [51]. The territory of the regional city mainly consists of mountains and hills, accounting for more than 80% of the total area. River valleys and basins are covered with dense forests [52]. The sea area of Fujian, reaching 136,000 square kilometers, is slightly larger than the land. The coastline is long and winding, with unique harbor resources. The weather in Fujian is mostly the subtropical marine monsoon climate, which is warm and humid. Fujian’s forest overage rate has ranked first in the country for 40 years consecutively. It reached as high as 65.95% in 2017, thanks to its superior geographic location and natural conditions [51].

In recent years, the economy in Fujian has developed rapidly. As shown in Table 1, Gross Domestic Product (GDP) increased from 0.45 trillion yuan in 2002 to 3.22 trillion yuan in 2017, with an annual growth rate of 14% [51,53]. Economic development has promoted the improvement of people’s living standards. From 2002 to 2017, the per capita disposable income of urban residents increased from 9189 yuan to 39,001 yuan, multiplied by nearly three times [51,53]. The per capita disposable income of rural residents increased from 3539 yuan to 16,335 yuan, multiplied by 3.6 times [51,53]. With economic development, the urbanization process is also steadily advancing. In 2002, the urban developed area was about 502.58 square kilometers, while increasing to 1516.88 square kilometers by 2017 [51,53]. The urban area has been swallowing huge amounts of land for agricultural and environmental uses. The scarcity of available land resources has gradually limited the development of urbanization. With the implementation of the “One Belt One Road” initiative in China and the deepening of economic globalization, Fujian has gained an increasingly prominent status. Consequently, it is of great significance to strengthen the research on the relationship between the urbanization in Fujian and the land use benefits. It is also necessary to explore the temporal and spatial evolution of its coordinated development proposing feasible plans. In this way, it is possible for us to promote the coordinated development and comprehensive management of cities in Fujian. Besides, we are also likely to avoid human-land conflicts and problems brought by urbanization. Finally, we can create a city cluster that is reasonably and scientifically organized.

### 2.2. Data Source and Processing

In view of data use, previous studies often used prefecture-level municipal districts data. However, it was difficult to accurately reflect the relationship between local land use benefits and the urbanization level. Moreover, compared with data of county-level cities and prefecture-level municipal districts, those of counties are incomplete. Consequently, we finally leverage the data of county-level cities and prefecture-level municipal districts in Fujian for this research. In terms of time, due to the missing data in some years, we select 2002, 2005, 2010, 2015, and 2017 as the time points for valuation. Among them, 2005, 2010, and 2015 are the end year of China’s 10th, 11th, and 12th Five-year Plan, respectively. They are of significance to the whole Five-year Plan. Besides, 2002 and 2017 are the earliest and latest data we can obtain. Many changes took place in some administrative regions within the study’s time interval. These changes include the abolishing of Jianyang City and the establishment of the Jianyang District of Nanping City in 2014, as well as the abolishing of Changle City and the establishment of Changle District of Fuzhou City in 2017 [51]. Consequently, the number of research objects were also changed. That is to say, there were 23 research objects in 2002, 2005, and 2010, 22 in 2015, and 21 in 2017. Considering this, we adjust the data accordingly. The data mainly come from the “Fujian Statistical Yearbook in 2003, 2006, 2011, 2015, 2017”, “China City Statistical Yearbook in 2003, 2006, 2011, 2015, 2017”. Note: China’s yearbooks record the previous year’s data, for example, “Fujian Statistical Yearbook 2003” records the data of Fujian Province in 2002.Missing data are supplemented by the statistical bulletin and statistical yearbook on the official website of the statistical bureau of each city.

To mitigate the impact of different dimensions or magnitude orders of indices, we first perform dimensionless processing to the data. We divide the indices into two categories, positive index and negative index according to their effect on the system [54]. Then, we standardize the data by Equation (1).
(1)$$x_{stk}=\{\begin{array}{c} \left(X_{stk}-m_{tk}\right)/\left(M_{tk}-m_{tk}\right)\begin{array}{cccc} & & & \left(Positive\;indicator\right) \end{array}\\ \left(M_{tk}-X_{stk}\right)/\left(M_{tk}-m_{tk}\right)\begin{array}{cccc} & & & \left(Negative\;indicator\right) \end{array} \end{array}$$

The $X_{stk}$ is the value and $x_{stk}$ is the standardized value. s is the number of regions, ranging from 1 to 23 (23 represents 2002, 2005, 2010; 22 represents 2015; and 23 represents 2017, respectively). t is the year, ranging from 1 to 5, representing 2002, 2005, 2010, 2015, and 2017, respectively. k is the number of indexes, ranging from 1 to 22. $m_{tk}$ refers to the minimum value of the k-th index in the t-th year of all regions, while $M_{tk}$ refers to the maximum value of the *k*-th index in the *t*-th year of all regions. Except for indices of urban population density, the rest indices are positive ones. Besides, all indices fall in the interval [0,1] after standardization.

### 2.3. Building the Index System of Urban Land Use Benefits and Urbanization Level

Constructing a scientific index evaluation system is a prerequisite for evaluating the coupling degree of urban land use benefits and urbanization [55]. To ensure scientificity, integrity, hierarchy, and operability, this research, based on the idea of synergy in physics and previous research, constructs a comprehensive evaluation index system of urban land use benefits and urbanization in Fujian. Urban land benefits are the four in one of the economic, social, ecological, and environmental benefits [14]. To comprehensively reflect this connotation, instead of only focusing on economic benefits, we evaluate urban land use benefits from the four dimensions of economic, social, ecological, and environmental benefits. Urbanization has a multidimensional meaning. It mainly includes the four interacting aspects of population migration, economic development, spatial expansion, and improvement of living standards [56]. Therefore, we perform a comprehensive evaluation of the four dimensions of population, economic, social, and spatial urbanization. In this way, we can avoid the one-dimensional view of the spatial transfer of the rural population that the government often thinks.

Specifically, to the index layer, the economic benefits of urban land use refer to the value of products and services produced per unit area of land. They can be measured by GDP, industrial output value, and fixed asset input [40]. Social benefits are mainly measured by population density, urban road area per capita, and developed area per capita [3]. For ecological benefits, the per capita park area, as well as the green coverage and area rate of the developed area, can fully reflect the urban ecological status. For environmental benefits, we usually use the pollutant compliance and removal rate for reflection. Due to the defects of statistical work in the statistical department, a large number of indices, such as industrial waste-water compliance rate and industrial solid comprehensive utilization rate, are missing. Consequently, we can only leverage the sewage and harmless treatment rate of domestic garbage to measure environmental benefits [57]. In the urbanization system, population urbanization refers to the process of transforming the agricultural population into the urban ones. Therefore, we leverage the rate of population urbanization and the number of nonagricultural populations to reflect the process of population urbanization [40]. In the process of urbanization, the economy is accumulating in the secondary industry and the tertiary one. Besides, the proportion of the tertiary industry keeps expanding. Therefore, we select per capita GDP, per capita industrial production value, and the proportion of tertiary industry to reflect economic urbanization [58]. In addition, social urbanization is a process that public service facilities continue to improve. Consequently, we choose indices, such as the number of hospital beds per 10,000 people, buses per 10,000 people, ordinary teachers per 10,000 people, and the total wages of urban employees, to reflect the process of social urbanization. We also reflect the process from multiple angles such as medical treatment, transportation, education, and wage level. The spatial urbanization is represented by the expansion of urban construction land, which is measured by urban construction land area and proportion of construction land [59].

Finally, we determined 8 primary indices and 22 secondary indices, as shown in Table 2.

### 2.4. Index Weighting

The weighting of indices is the core of multiattribute decision-making. The weighting method mainly includes subjective and objective types. The subjective weighting method mainly weights by expert experience. However, this method is likely to be interfered with by subjective factors. Besides, it also does not fully utilize the information in the data. Comparatively, the core idea of the objective one is to give indices weight by comparing the content or differentiation of the data in indices. The more obvious the change in a certain index is, the richer the information the index contains, and the heavier the weight is [60]. Overall, the objective weighting method is more appropriate. Nevertheless, it is also a problem to choose the most suitable one from various objective weighting methods.

The Gini coefficient is an important analysis index in economics. It is internationally leveraged to comprehensively measure the difference in the distribution of income among residents. Its method of calculation coincides with the core idea of the objective weighting method. For both, the greater the degree of data differentiation is, the larger the value is. The Gini coefficient weighting method draws on this method, weighting each index by deriving the Gini coefficient value. It comprehensively reflects the difference between any two data in the same index. It fully utilizes the value of information. It scientifically and objectively reflects the differentiation (distinction) of the data in a certain index. In addition, the definition of the Gini coefficient itself eliminates the effect of dimensions. Consequently, there is no need for us to standardize data in advance. As a result, it avoids the loss of information in data processing. It simplifies the calculation. Besides, its applicability is stronger and order preservation is better, compared to other objective weighting methods [61]. Therefore, we leverage the Gini coefficient weighting method in our index system. The steps are as follows [61]:
Calculate the Gini coefficient value of the evaluation indices, as shown in Equations (2) and (3):
(2)$$G_{k}=\sum_{i=1}^{n}\sum_{j=1}^{n}\frac{\left|X_{ki}-X_{kj}\right|}{2n^{2}\mu_{k}}\;(k\;=1,2,\ldots ,m\;\mathrm{and}\;\mu \neq 0)$$
(3)$$G_{k}=\sum_{i=1}^{n}\sum_{j=1}^{n}\frac{\left|X_{ki}-X_{kj}\right|}{n^{2}-n}\;(k\;=1,2,\ldots ,m\;\mathrm{and}\;\mu =0)$$
where $G_{k}$ is the Gini coefficient value of the k-th index. m is the number of evaluation indices. n is the number of samples of the indices. $X_{ki}$ refers to the i-th sample of the k-th index. $\mu_{k}$ refers to the sample of the k-th index. In particular, when the average value of the index data a not 0, the Gini coefficient value is calculated by Equation (2). When it is 0, the value is calculated by Equation (3).

2.Calculate the Gini coefficient weight of the evaluation index:

After calculating the Gini coefficient values $G_{k}$ of m indices in the evaluation system by Equation (2) or (3), and then normalizing them, the Gini coefficient weight $g_{k}$ of the k-th index can be derived, as shown in Equation (4):(4)$$g_{k}=\frac{G_{k}}{\sum_{i=1}^{m}G_{i}},$$
where $g_{k}$ is the weight of the Gini coefficient of the k-th index. $G_{k}$ is the value of the Gini coefficient of the k-th index. m is the number of evaluation indices. The sum of all weights is 1. The specific weight values are shown in Table 2.

### 2.5. Relative Development Model

The relative development degree is a specific index that reflects the urban land use benefits as well as the relative development degree and status of urbanization in a certain period. The calculation steps are as follows [58].

Calculate the overall benefits of urban land use benefits and urbanization system, as shown in Equations (5) and (6):(5)$$f_{st}\left(x\right)=\sum_{k=1}^{m_{1}}a_{k}x_{stk}\;(s=1,2,\ldots ,n_{1}\;\mathrm{and}\;t=1,2,\ldots ,n_{2}),$$
(6)$$g_{st}\left(y\right)=\sum_{k=1}^{m_{2}}b_{k}y_{stk}\;(s=1,2,\ldots ,n_{1}\;\mathrm{and}\;t=1,2,\ldots ,n_{2}),$$

Among them, $f_{st}\left(x\right)$ and $g_{st}\left(y\right)$ are the overall benefits of the urban land use benefits and urbanization system in the t-th year in the s area. $n_{1}$ and $n_{2}$ refer to the sample sizes in areas and years, respectively. $x_{stk}$ and $y_{stk}$ refer to the standardized values of the k-th index in the t-th year in the s-th area, respectively. $a_{k}$ and $b_{k}$ refer to the weights of the corresponding indices, respectively. $m_{1}$ and $m_{2}$ refer to the numbers of indices in the urban land use benefits and urbanization system, respectively. The two are added as the indices in the evaluation system, that is, $m_{1}+m_{2}=m$.

2.Calculate the relative development degree, as shown in Equation (7):(7)$$E_{st}=\frac{f_{st}\left(x\right)}{g_{st}\left(y\right)}\;(s=1,2,\ldots ,n_{1}\;\mathrm{and}\;t=1,2,\ldots ,n_{2}),$$
where $E_{st}$ is the relative development degree of urban land use benefits and urbanization in the s-th area in the t-th year. When $E_{st}>1$, we call it an advanced city; when $E_{st}=1$, we call it a synchronous city; when $E_{st}<1$, we call it a lagging city.

### 2.6. Coupling Coordination Degree (CCD) Model

Refer to Tang’s studies, we build the CCD model by the following steps [62], as shown in Equations (8)–(10):(8)$$C_{st}\left(x,y\right)=2\sqrt{\frac{f_{st}\left(x\right)\times g_{st}\left(y\right)}{{\left[f_{st}\left(x\right)+g_{st}\left(y\right)\right]}^{2}}}\;(s=1,2,\ldots ,n_{1}\;\mathrm{and}\;t=1,2,\ldots ,n_{2}),$$
(9)$$T_{st}\left(x,y\right)=\alpha f_{st}\left(x\right)+\beta g_{st}\left(y\right)\;(s=1,2,\ldots ,n_{1}\;\mathrm{and}\;t=1,2,\ldots ,n_{2}),$$
(10)$$D_{st}\left(x,y\right)=\sqrt{C_{st}\times T_{st}}\;(s=1,2,\ldots ,n_{1}\;\mathrm{and}\;t=1,2,\ldots ,n_{2}),$$

Among them, $C_{st}$ is the coupling degree of urban land use benefits and urbanization of the s-th area in the t-th year, and $C_{st}\in \left[0,1\right]$. $T_{st}$ is the coordination degree of urban land use benefits and urbanization of the s-th area in the t-th year. $D_{st}$ is the coupling coordination degree of urban land use benefits and urbanization of the s-th area in the t-th year, and $D_{st}\in \left[0,1\right]$. α and β are the contributions of the urban land use benefits system and the urbanization system, respectively. According to the existing literature, we cannot conclude whether the urban land use benefits and the urbanization are more important, so we take 0.5 for both α and β. According to the existing research, we divide the CCD into three stages and 10 categories [63], as shown in Table 3.

## 3. Results

### 3.1. Weight Analysis

It can be seen from Table 2 that the difference between the weights of the secondary indices is huge. The largest is 0.1084, while the smallest is 0.0062, 17.48 times as large as the latter. Among the 22 secondary indices, there are three indices, concentrated on economic benefits and spatial urbanization, which are above 0.1. There are five indices between 0.05 and 0.099. Fourteen are below 0.05. As shown in Table 4, among the primary indices, the weight of the economic benefit index is the largest, close to 1/3 of the total. Besides, the difference between the weight of the ecological benefits index and the environmental benefits index is not large, far lower than the weights of other primary indices. To conclude, it shows that the gap between ecological and environmental benefits among regions is narrow. However, the gap between economic benefits and spatial urbanization is huge, comparatively.

### 3.2. Relative Development Analysis

From Figure 2, in the five years of 2002, 2005, 2010, 2015, and 2017, the relative development degree of nine prefecture-level municipal districts and 14 county-level cities in Fujian Province did not take 1. This shows that the urban land use benefits and urbanization in various regions did not develop synchronously in each year. Considering the overall trend, the type of mainstream cities has transferred from advanced cities to lagging ones. From 2002 to 2017, the number of advanced cities was 19, 15, 12, 12, and 10, respectively, with the proportions of 82.6%, 65.2%, 52.2%, 54.5%, and 47.6%. This situation is more obvious in prefecture-level municipal districts. The number of lagging prefecture-level municipal districts was two in 2002, while reducing to 0 by 2017. Among them, Fuzhou and Xiamen were lagging cities in the past five years.

The relative development degree is derived by the ratio of the urban land use benefits and the urbanization level. Considering this, we cannot help asking whether urban land use benefits are lower in areas with a low relative development degree. Taking 2017 as an example, we make a comparison chart of urban land use benefits and urbanization level in various regions, as shown in Figure 3.

From Figure 3, the urban land use benefits in areas with relatively low development are higher. Correspondingly, the urbanization level is also higher. From the perspective of the administrative level, the urban land use benefits and urbanization level of prefecture-level municipal districts, such as Fuzhou, Xiamen, Zhangzhou, Quanzhou, Sanming, and Longyan, are generally higher than their county-level cities. From the perspective of the regional economic development level, Fuzhou, Xiamen, Quanzhou, Zhangzhou, and other well-developed regions have higher levels of urbanization. However, the discrepancy of land use benefits between well-developed regions and others is much narrower than that of urbanization levels.

### 3.3. Analysis of the Coupling Coordination Degree (CCD) of Urban Land Use Benefits and Urbanization Level

From Table 5, we can see that in the five years of 2002, 2005, 2010, 2015, and 2017, regions whose coupling degree C of urban land use benefits and urbanization level were above 0.9 accounted for 100%, 100%, 95.6%, 95%, and 100%, respectively. Only ratios of Xiamen in 2010 and Shaowu in 2015 were below 0.9, while still between 0.8 and 0.9. This variable is relatively stable. The coupling coordination degree D was between 0.2 and 0.6. The change is relatively obvious. From Figure 4, it can be seen that the coupling coordination degree is mainly in four stages. They are moderately uncoordinated, slightly uncoordinated, at the edge of being uncoordinated, and barely coordinated. It is moderately uncoordinated and slightly uncoordinated that dominate in them. These two stages accounted for 74%, 74%, 82.6%, 68.2%, and 66.7% of the previous years, respectively. No region has reached the stage of coordinated development. However, most have moved from moderately uncoordinated to slightly uncoordinated, with an overall trend to develop well. Considering the stage of each region, there are obvious trends and laws of distribution.

In view of the time series, from 2010 to 2015, there were eight cities that achieved the improvement of the coupling coordination phase. The number far exceeded the total of other adjacent years. In addition, only the coupling coordination phase in one city had a reduction. This number of reductions was also far lower than that in other adjacent years. From the perspective of spatial distribution, regions with relatively high coupling coordination, such as Xiamen, Fuzhou, Quanzhou, and Shishi, are all coastal regions. Except for Sanming, it is an inland city. Moreover, Xiamen and Fuzhou become the coastal central cities while Sanming is the inland central city. Consequently, there grows a sort of “central-peripheral” development pattern. That is, for the surrounding cities of these three central cities, the closer to the central city they are, the higher their coordination degree is, and vice versa.

## 4. Discussion

The type of mainstream cities in Fujian Province has changed from leading cities to lagging ones. It reflects that in the urbanization process, the development of urban land use benefits lags behind urbanization. This also confirms that the rapid development of urbanization has brought many defects, such as the spreading development of urbanization and inefficient land use [64,65]. The main reason is that, in the process of urbanization, local governments blindly expand urban areas. They overpursue urbanization while neglecting the efficient utilization of land. The “BIG PIE” policy has resulted in low urban land use benefits, far behind the development of urbanization. Moreover, the deeper reason is related to the assessment mechanism of government officials. The assessment mainly considers economic indices. Since urban land expansion has a motivating effect on economic development [66,67], government officials are encouraged to expand urban areas. It causes that officials only focus on socioeconomic benefits in land planning, while ignoring the ecological and environmental benefits. As a result, it finally leads to the fact that urban land use benefits are lower than the urbanization level.

Besides, we find that the urban land use benefits and urbanization level are related to the administrative level and economic development of the city. It is manifested that areas with higher administrative and economic development levels tend to own higher urban land use efficiency and urbanization level. However, the impact of economic conditions on urbanization is greater than that of the urban land use benefits. The reason is not complex to explain. One the one hand, areas with high administrative level will receive more policy support and resource tilt [68]. As a result, it is more beneficial to urban construction and economic development. Comparatively, economically well-developed areas tend to pay more attention to economic and social benefits. Nevertheless, economic development often comes at the price of the environment. This partially hinders the improvement of overall benefits. Consequently, the difference in urban land use benefits among cities is insignificant.

Moreover, it is also found that the CCD is still at a relatively low level in various regions of Fujian. However, it develops in a good direction. There are situations of high coupling and low coupling coordination. This is mainly because the coupling degree only indicates the strength of the effect between urban land use benefits and the urbanization level. However, we do not know whether they promote or inhibit each other. Comparatively, the coupling coordination degree fully considers the coordination degree between them [69]. Early urban land use benefits limited to urbanization. However, the two gradually promote each other as time goes by. This situation does not only occur in Fujian exclusively. Jia et al. have studied the three major urban agglomerations, including the Yangtze River Delta, the Pearl River Delta, and the Beijing–Tianjin–Hebei area, which are relatively better-developed [70]. They obtained the same result. Similarly, Zuo et al. also got the same result when studying the Shaanxi–Gansu-Ningxia region, which is of lower development level [40]. It requires further study of whether this situation is suitable for the whole country. However, the increase in the CCD took place in large numbers from 2010 to 2015. It far exceeded the sum of other years. This may partly benefit from the establishment of the Western Taiwan Straits Economic Zone in 2009 and the peaceful cross-strait relations.

Finally, we find that each city generally exhibits a development pattern according to its distance from the three central cities of Fuzhou, Xiamen, and Sanming. That is, the closer to the central city it is, the higher the coordination degree it has; vice versa. What is more, cities with developed coastal transportation have a higher coupling and coordination degree than inland cities. Zhang et al. [3] and Wang et al. [58] studied the three provinces in Northeast China and the Bohai rim area, respectively, to obtain this “center-periphery” pattern. Zhang et al. found that cities with relatively high coupling and coordination degrees are distributed along the Harbin–Dalian Transportation Economic Belt in strips [3]. This finding is so interesting. It shows that the development of regions not only depends on their own influencing factors, but also on the radiation and leading role of the central cities. The farther away the distance is, the weaker this radiation and driving effect is. Meanwhile, developed traffic can also strengthen this driving effect. This provides a way for the government to give full play to the leading role of central cities. However, there are still several questions that require us to perform further research and practice. For example, we should consider how to leverage the central city’s radiation and driving role more efficiently. This is significant to promote the coordinated development of land use benefits and urbanization in other cities.

When it comes to research methods, the research methods of this study are applicable to various areas with similar issues, such as urban agglomerations, provinces, and cities. However, the completeness of the data in each region, the difference of regional conditions, and the different perspectives of the target connotation will all affect the construction of the index system. The universal system still needs to be tested in practice. Besides, the selection of the index weighting method will also have an impact on the results. Weights, computed by the objective weighting method, are based on the information of the data itself [61]. Due to the discrepancy in the data of various regions, weights will change accordingly. Consequently, weights obtained in this study cannot be directly applied to other regions.

In addition, this research also has several drawbacks, which are likely to be solved in future research. (1) This study only selects 2002, 2005, 2010, 2015, and 2017 as time nodes, causing the time series data to be insufficient. Consequently, we can only tentatively study the spatial and temporal evolution of urban land use benefits and urbanization in various regions. Therefore, future research can leverage continuous-time data to analyze its evolution in more detail. (2) Considering a large number of county statistical data is missing, we only take prefecture-level municipal districts and county-level cities as research samples. However, the county is also an important part of China’s urbanization process. As a result, there are limitations in the spatial analysis of urban land use benefits and urbanization in regions of Fujian; (3) In the construction of the index system, some data are missing. For example, compared with other studies, some important indices, such as the proportion of the tertiary industry’s employed population [59], urban residents per capita use area [57], and the three-waste treatment rate [54], are not included. Consequently, it requires us to add more comprehensive data in future relevant studies to improve the index system, reflecting the spatiotemporal evolution of the CCD of each city more accurately.

## 5. Conclusions

It is of practical significance to study the law of spatiotemporal evolution of the coupling and coordination relation between urban land use benefits and urbanization level in Fujian. It is likely to promote the healthy development of the urbanization process. Therefore, this study takes county-level cities and prefecture-level municipal districts in Fujian Province as the research object, leveraging the data of 2002, 2005, 2010, 2015, and 2017 to construct the evaluation index system of urban land use benefits and urbanization system. We derive the CCD of the index system and analyze the temporal and spatial evolution law. The conclusions of this study are as follows:
(1)Urban land use benefits and urbanization levels are positively correlated with the regional administrative level and economic development status.(2)The CCD of urban land use benefits and urbanization levels in various regions of Fujian is still low. However, the overall development direction is good.(3)In terms of spatial distribution, the CCD has a “center-periphery” pattern. That is, the closer to the three central cities of Fuzhou, Xiamen, and Sanming it is, the higher the coordination degree it has; vice versa.

The above findings can provide certain implications for policy formulation. First of all, in the process of urbanization, Fujian Province cannot blindly pursue the speed. It is necessary to plan urban expansion rationally and strengthen the intensive use of land, avoiding the development model of “BIG PIE”. It is also significant to add some indices on the ecological environment to the assessment of government officials. It can avoid only concentrating on the economy while neglecting the ecological environment. It can finally promote the overall improvement of urban land use benefits. Secondly, it is necessary to give targeted priority to the improvement of the urban land use benefits or urbanization level, based on the reality of different regions. Finally, it is recommendable to strengthen the radiation and the leading role of central cities to surrounding ones. Appropriate policy and resource support for peripheral cities should be provided, such as financial subsidies for regional construction.

## Figures and Tables

**Figure 1 ijerph-17-05647-f001:**
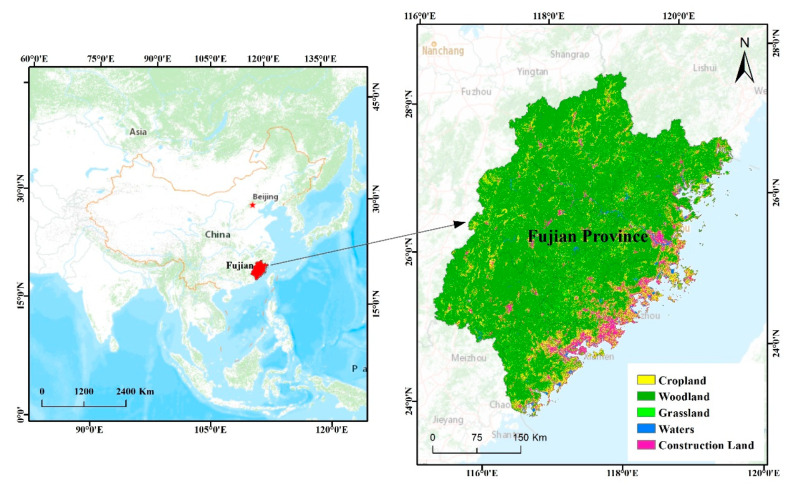
The location of the study area.

**Figure 2 ijerph-17-05647-f002:**
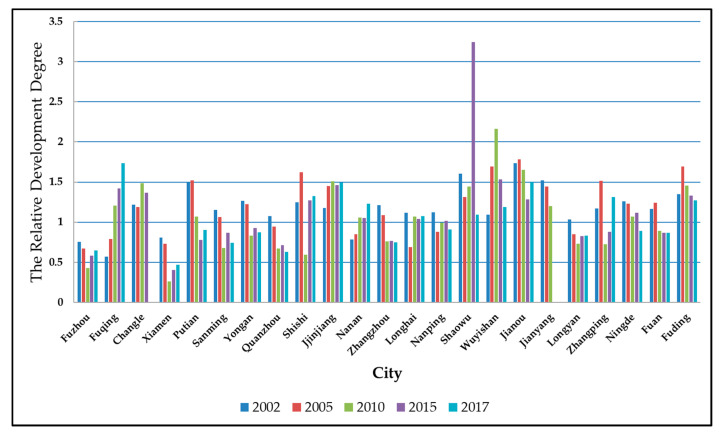
The relative development degree of urban land use efficiency and urbanization level.

**Figure 3 ijerph-17-05647-f003:**
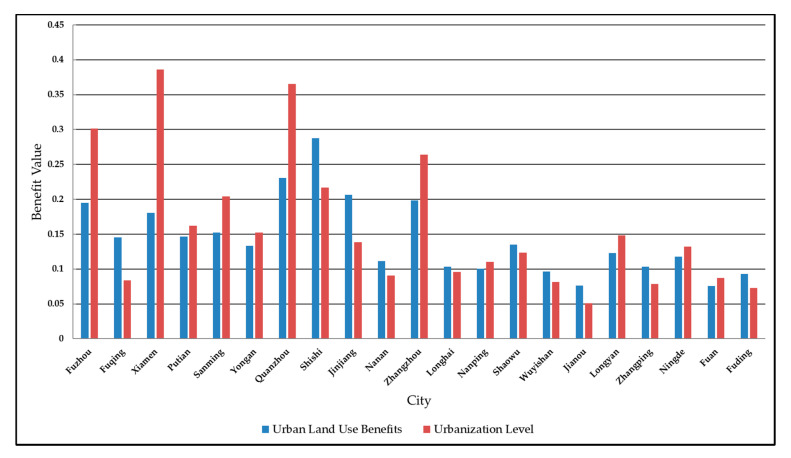
Comprehensive value of urban land use benefits and urbanization level by region in 2017.

**Figure 4 ijerph-17-05647-f004:**
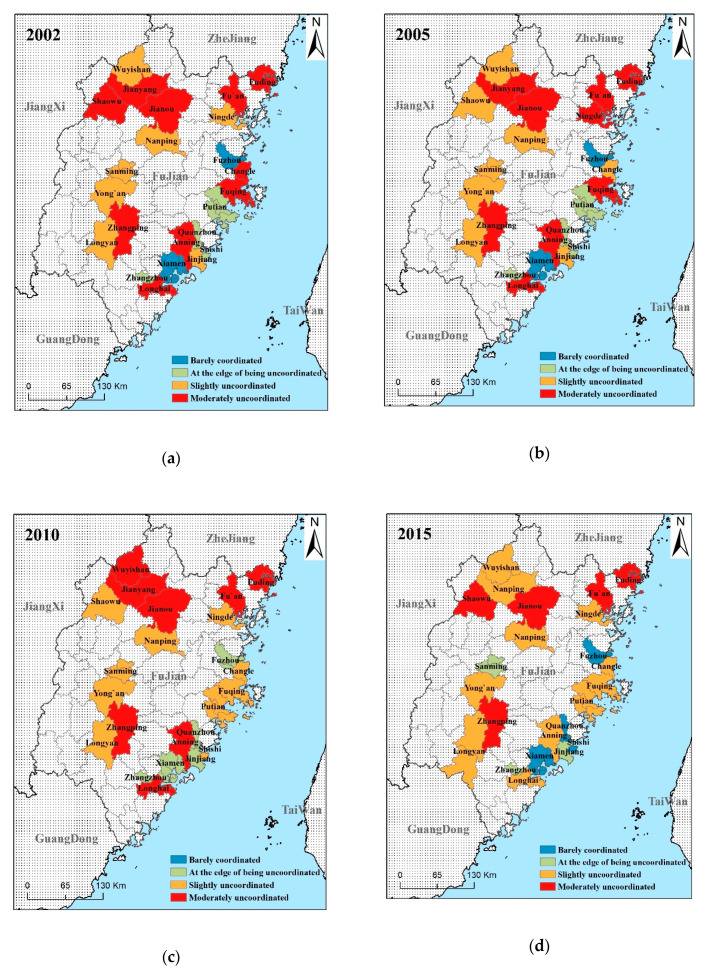
The spatial distribution diagram of the regional coupling coordination degree. (**a**)—The spatial distribution diagram of coupling coordination degree in 2002.; (**b**)—The spatial distribution diagram of coupling coordination degree in 2005. (**c**)—The spatial distribution diagram of coupling coordination degree in 2010.; (**d**)—The spatial distribution diagram of coupling coordination degree in 2015.; (**e**)—The spatial distribution diagram of coupling coordination degree in 2017.; (**f**)—The spatial distribution diagram of Variation type of coupling coordination degree.

**Table 1 ijerph-17-05647-t001:** Basic economic information of Fujian Province.

Year	GDP (Trillion yuan)	PCDIUR (yuan)	PCDIRR (yuan)
2002	0.45	9189	3539
2005	0.66	12,321	4450
2010	1.47	21,781	7427
2015	2.6	33,275	13,793
2017	3.22	39,001	16,335

GDP—Gross Domestic Product; PCDIUR—the Per Capita Disposable Income of Urban Residents; PCDIRR—the Per Capita Disposable Income of Rural Residents.

**Table 2 ijerph-17-05647-t002:** Evaluation index system of urban land use benefits and urbanization level.

Item	Primary Index	Secondary Index	Weight
Urban land use benefits	Economic benefits	GDP per unit area (CNY 10,000)	0.1050
		Investment in fixed assets per unit area (CNY 10,000)	0.0998
		gross industrial output value per unit area (CNY 10,000)	0.1084
	Social benefits	urban population density (person /km^2^)	0.0657
		Urban road area per capita (m^2^/person)	0.0258
		Developed area per capita (m^2^/person)	0.0479
	Ecological benefits	Park area per capita (m^2^/person)	0.0156
		Developed coverage rate in the built-up area (%)	0.0062
		urban green land rat (%)	0.0078
	Environmental benefits	Sewage treatment rate (%)	0.0181
		Harmless treatment rate of domestic garbage (%)	0.0087
Urbanization level	Economic urbanization	GDP per capita (1000 yuan/person)	0.0289
		Industrial production value per capita (1000 yuan/person)	0.0426
		Proportion of tertiary industry (%)	0.0183
	Social urbanization	Number of hospital beds per 10,000 people (per 10,000 people)	0.0378
		Number of buses per 10,000 people (vehicles/10,000 people)	0.0535
		Total wages of urban employees on the job (yuan)	0.0101
		Number of ordinary teachers per 10,000 people (people /10,000 people)	0.0154
	Population urbanization	Population urbanization rate (%)	0.0234
		Non-agricultural population (10,000 people)	0.0717
	Spatial urbanization	Urban construction land area (km^2^)	0.0859
		Proportion of construction land (%)	0.1034

GDP—Gross Domestic Product; CNY—China Yuan.

**Table 3 ijerph-17-05647-t003:** Discriminating standards of the coupling coordination degree.

Stage	*D* Value	Category
	0–0.099	Extremely uncoordinated
Uncoordinated development	0.10–0.199	Seriously uncoordinated
	0.20–0.299	Moderately uncoordinated
	0.30–0.399	Slightly uncoordinated
Transitional development	0.40–0.499	At the edge of being uncoordinated
	0.50–0.599	Barely coordinated
	0.60–0.699	Slightly coordinated
Coordinated development	0.70–0.799	Moderately coordinated
	0.80–0.899	Well-coordinated
	0.90–1.00	Perfectly coordinated

**Table 4 ijerph-17-05647-t004:** Weights of primary indices.

Primary Index	Weight
Economic benefits	0.3132
Social benefits	0.1394
Ecological benefits	0.0296
Environmental benefits	0.0268
Economic urbanization	0.0898
Social urbanization	0.1168
Population urbanization	0.0951
Spatial urbanization	0.1893

**Table 5 ijerph-17-05647-t005:** Coupling coordination degree (CCD) of urban land use benefits and urbanization level in various regions.

City	2002		2005		2010		2015		2017	
	C	D	C	D	C	D	C	D	C	D
Fuzhou	0.990	0.551	0.981	0.542	0.917	0.485	0.964	0.526	0.977	0.492
Fuqing	0.962	0.289	0.993	0.278	0.996	0.300	0.985	0.341	0.963	0.332
Changle	0.995	0.282	0.996	0.311	0.981	0.320	0.988	0.360		
Xiamen	0.994	0.539	0.988	0.532	0.814	0.427	0.907	0.513	0.932	0.514
Putian	0.980	0.487	0.978	0.495	0.999	0.331	0.992	0.394	0.999	0.392
Sanming	0.997	0.391	0.999	0.381	0.981	0.389	0.998	0.419	0.989	0.420
Yongan	0.993	0.333	0.995	0.335	0.996	0.354	0.999	0.372	0.998	0.377
Quanzhou	0.999	0.463	1.000	0.466	0.981	0.441	0.986	0.507	0.974	0.539
Shishi	0.994	0.402	0.971	0.418	0.967	0.367	0.993	0.503	0.990	0.500
Jinjiang	0.997	0.385	0.983	0.372	0.979	0.498	0.982	0.409	0.980	0.411
Nanan	0.993	0.239	0.997	0.238	1.000	0.287	1.000	0.327	0.995	0.317
Zhangzhou	0.995	0.405	0.999	0.450	0.991	0.407	0.991	0.474	0.990	0.478
Longhai	0.999	0.226	0.983	0.216	0.999	0.260	1.000	0.300	0.999	0.316
Nanping	0.998	0.335	0.998	0.337	1.000	0.334	1.000	0.333	0.999	0.324
Shaowu	0.973	0.300	0.991	0.321	0.983	0.338	0.849	0.262	0.999	0.359
Wuyishan	0.999	0.328	0.966	0.310	0.930	0.288	0.977	0.315	0.996	0.298
Jianou	0.963	0.220	0.960	0.224	0.969	0.237	0.992	0.251	0.980	0.250
Jianyang	0.978	0.264	0.983	0.259	0.996	0.271				
Ningde	0.993	0.305	0.995	0.296	0.999	0.310	0.998	0.348	0.998	0.354
Fuan	0.997	0.222	0.994	0.239	0.998	0.256	0.997	0.292	0.998	0.285
Fuding	0.989	0.220	0.966	0.249	0.983	0.275	0.990	0.297	0.993	0.287
Longyan	1.000	0.391	0.997	0.382	0.988	0.388	0.995	0.362	0.996	0.367
Zhangping	0.997	0.250	0.979	0.239	0.988	0.272	0.998	0.281	0.991	0.300

C—the Coupling Degree of Urban Land Use Benefits and Urbanization; D—the Coupling Coordination Degree of Urban Land Use Benefits and Urbanization.

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
