# Peer review of "The Coupling and Coordinated Development from Urban Land Using Benefits and Urbanization Level: Case Study from Fujian Province (China)"

_ijerph, 2020, doi:10.3390/ijerph17165647_

Round 1
Reviewer 1 Report
This paper takes problem of the evaluation of urban land using benefits and urbanization level in fast urbanization area in China. The proposed methodology with statistical data are approbated and often used in similar studies. Paper structure is correct and the obtained results are well argued. Below I indicate mainly problems with you article.
General comments:
1/ Title can be simplify to: ‘The Coupling and Coordinated Development from Urban Land Using Benefits and Urbanization Level: case study from Fujian Province (China)’.
2/ Abstract needs correction with more exposing of resorach aims, methods and main results.
2/ Introduction needs correction and rebuilding according to: (1) research aim and reason of investigation and 2) review of the state of art of coupling urban land use benefits and urbanization level. I propose an integration of two chapters: Introduction and Literature review to one.
3/ Data and Methodology need supplement. I propose also to add the Table with basic social-economical information of cities and regions.
4/ Results section. In my opinion you can to put the tables with calculation results, e.g. for evaluating index system (22 indicies), the Gini coefficient, specific weights from Table 1,
5/ In Discussion section you can show the discussion your obtained results with published similar studies.
6/ Conclusions need correction. I think that one paragraph with showed results you can to move to Discussion section – it help you to preparation of discussion.
Detailed comments are provided in the text (enclosed pdf).

Author Response
Dear Reviewer:
We appreciate your detailed comments, which have greatly helped us polish our paper. We feel sorry for the defects in the previous draft submitted. We have modified our paper accordingly, as is shown in the paper with red text.
We answer your concerns as follows:
1/ Title can be simplified to: ‘The Coupling and Coordinated Development from Urban Land Using Benefits and Urbanization Level: case study from Fujian Province (China)’.
We think the title you wrote is perfectly fit our research. Thus, we decide change our title to: ‘The Coupling and Coordinated Development from Urban Land Using Benefits and Urbanization Level: case study from Fujian Province (China)’
2/ Abstract needs correction with more exposing of research aims, methods and main results.
We rewrite the abstract and add more research aims, methods and main results.
3/ Introduction needs correction and rebuilding according to: (1) research aim and reason of investigation and 2) review of the state of art of coupling urban land use benefits and urbanization level. I propose an integration of two chapters: Introduction and Literature review to one.
We merge the introduction and literature review together and delete some parts that not directly related to the research. We also adjust the rest parts with your suggestion.
4/ Data and Methodology need supplement. I propose also to add the Table with basic social-economical information of cities and regions.
We add the table with basic social-economical information of cities and regions to make the article more reliable.
5/ Results section. In my opinion you can to put the tables with calculation results, e.g. for evaluating index system (22 indicies), the Gini coefficient, specific weights from Table 1,
The specific weights we showed already cover all information of the Gini coefficient which act as an intermediate variable during the calculation. Thus, we only describe the weights in our results. Moreover, we list the wights of the first level indicators because table 2 already represent all 22 secondary indicators.
6/ In Discussion section you can show the discussion your obtained results with published similar studies.
We rewrite the discussion parts and add related literature as you suggest.
7/ Conclusions need correction. I think that one paragraph with showed results you can to move to Discussion section – it help you to preparation of discussion.
We do not make a significant change of the conclusion parts according to the opinions from other reviewers. However, we list the results when we rewrite the discussion and improve the structure to increase readability.
8/How id different between ecological and environmental benefits?
Ecological benefits are more about biology such as plants, and environmental benefits are more about non-biology parts such as water resources. We show the difference in table 2.
Based on other reviewers’ advice, we have modified the paper as follows:
- We merge figure 1 and figure 2 together and remake figure 3,4 & 5 to present more information;
- The introduce of coupling degree and coupling coordinated degree would be found in introduction L70-78;
- We redescribe the data parts;
- The sum of the weights showed in table 1 (original table 2) is equal to 1, the supplementary description would be found in L260-261;
- We format the formulas and characters to increase the readability;
- We correct all grammar problems and add more reference about the research region.
Again, we appreciate your time reviewing our paper and your comments which have been valuable to us. We would love to hear from you should there be any further questions or advice.
Best regards,
Authors
Reviewer 2 Report
Thank you for the opportunity to read the article.
I have some comments on your article.
The title of the article is very long, think about shortening.
ABSTRACT
The abstract must contain the basic idea, question what you want to research. The abstract must contain the main result of the research. Your index has name? So named it. For readers that are not familiar with CCD and method of coupling the abstract is unclear.
INTRODUCTION
The explanation of "Coupling" and "Coordination degree" are at 9th page. For readers, it is very late. Shortly present it in the introduction.
row 88 "... background, This..." -??
row 88 - 93 - the sentences are the same as in Abstact
Chapter 2
row 105 "... geographer Alonso W studied..." - remove the W or change to W.
Chapter 3
row 182. use delimiter in numbers like 9,189 yuan, respectively 39,001 yuan.
row 202, 203 - firstly mentioned "research subjects", followed by "research objects". What are the objects - district/regions?
row 211 - 215 write the variables in text with subscripts. There are maybe pictures in the text and it is very disturbing.
row 280- 290 - the same problem with writing variables from equations like Gk in text. Use italics for all variables like "k" etc.
Table 1 - For unit m2 use the text font. There are three strange fonts or pictures? The same km2 at the bottom.
The sum of Weights are 1 in Table 1? If yes, please, mention it in text.
"10000 people" - use delimiter like "10,000 people" in all other cases
Chapter 4
row 373 - Figure X ?
Table 3 is very wide and long. Please highlight the values that are mentioned in the text below e.g. by the bold font in the table. It increases the readability of the table.
Figure 5 - The maps are not correct form the cartographical point. Firstly, the level of detail of borders is very detail for used scale. E.g. islands etc and course of lines must be smoothed. Use proper map data for used scale with a higher level of cartographic generalization. Add to the map borderline where are other lands (regions) - on the west maybe. From the presented map is not evident where is the sea and where is earth. For readers from other countries is hard to join information from the text about "coastal regions".
The brown and dark blue color has very high color saturation and, subsequently, the text labels with the names above the polygons are not readable. For the labels and legend use sanserif font.
Concerning the maps. It will be beneficial to show one more final map that will express the trend of change for all periods in one map. E.g some district remains stable, some have a positive trend and some cyclic trend. It will be interesting to follow the trend in districts. In that case must be very clear selection of colors for aech category (e.g. 1. "stable - barely coor." 2. "stable - at the edge ..", 3. "stable slightly uncoor." etc. ; 5. change from "barely to at the edge .." ,....)
DISCUSSION, CONCLUSION
I have a question. Is it possible to use methodology, particularly the same sugested weights in Table 1 for another province in China? Please, mention it in the Discussion of Conclusion.
Author Response
Dear Reviewer:
We appreciate your detailed comments, which have greatly helped us polish our paper. We feel sorry for the defects in the previous draft submitted. We have modified our paper accordingly, as is shown in the paper with red text.
We answer your concerns as follows:
- ABSTRACT:The abstract must contain the basic idea, question what you want to research. The abstract must contain the main result of the research. Your index has name? So named it. For readers that are not familiar with CCD and method of coupling the abstract is unclear.
We rewrite abstract follow your suggestion and make sure the new one contains the points you mentioned. However, the number of indexes is too large to list. We can only list the type of them. Moreover, the description of coupling coordination degree (CCD) model is over complicated for abstract. We add some simple idea about it here.
- INTRODUCTION: The explanation of "Coupling" and "Coordination degree" are at 9th page. For readers, it is very late. Shortly present it in the introduction.
row 88 "... background, This..." -??
row 88 - 93 - the sentences are the same as in Abstact
We describe coupling coordination degree (CCD) in L70-78 of introduction. According to your opinion, we deeply agree that we should mention it as earlier as possible for readers. We also merge the introduction and literature review together and delete some irrelevant parts.
- Chapter 2: row 105 "... geographer Alonso W studied..." - remove the W or change to W.
We delete this part because the rewriting of the literature review.
- Chapter 3: row 182. use delimiter in numbers like 9,189 yuan, respectively 39,001 yuan.
row 202, 203 - firstly mentioned "research subjects", followed by "research objects". What are the objects - district/regions?
row 211 - 215 write the variables in text with subscripts. There are maybe pictures in the text and it is very disturbing.
row 280- 290 - the same problem with writing variables from equations like Gk in text. Use italics for all variables like "k" etc
We format all formula and characters and add numerical separator. The expression of ‘research subjects’ and ‘research objects’ have been unified.
- Table 1: For unit m2 use the text font. There are three strange fonts or pictures? The same km2at the bottom.
The sum of Weights are 1 in Table 1? If yes, please, mention it in text.
"10000 people" - use delimiter like "10,000 people" in all other cases.
We format all symbols. The sum of the weights showed in table 1(original table 2) is equal to 1. The supplementary description would be found in L260-261.
- Chapter 4: row 373 - Figure X ?
Table 3 is very wide and long. Please highlight the values that are mentioned in the text below e.g. by the bold font in the table. It increases the readability of the table.
We correct the figure X which should be original figure 5(present figure 4). We do not reduce the information of the original table 3(present table 5) which is the foundation of the description following to present more details for readers.
- Figure 5 - The maps are not correct form the cartographical point. Firstly, the level of detail of borders is very detail for used scale. E.g. islands etc and course of lines must be smoothed. Use proper map data for used scale with a higher level of cartographic generalization. Add to the map borderline where are other lands (regions) - on the west maybe. From the presented map is not evident where is the sea and where is earth. For readers from other countries is hard to join information from the text about "coastal regions".
The brown and dark blue color has very high color saturation and, subsequently, the text labels with the names above the polygons are not readable. For the labels and legend use sanserif font.
Concerning the maps. It will be beneficial to show one more final map that will express the trend of change for all periods in one map. E.g some district remains stable, some have a positive trend and some cyclic trend. It will be interesting to follow the trend in districts. In that case must be very clear selection of colors for aech category (e.g. 1. "stable - barely coor." 2. "stable - at the edge ..", 3. "stable slightly uncoor." etc. ; 5. change from "barely to at the edge .." ,....)
We merge figure 1 and 2 together and remade other figures with more details as you suggested. We put a trend type chart in figure 4(original figure 5) to keep the figure clean and orderliness. We do not change the font because the sanserif font does not present well in figures.
- DISCUSSION, CONCLUSION:I have a question. Is it possible to use methodology, particularly the same suggested weights in Table 1 for another province in China? Please, mention it in the Discussion of Conclusion.
We discuss this in the fourth paragraph of the discussion and rewrite the conclusion.
Based on other reviewers’ advice, we have modified the paper as follows:
- We change our title to“The Coupling and Coordinated Development from Urban Land Using Benefits and Urbanization Level: case study from Fujian Province (China)” as other reviewers suggestedï¼›
- We correct all grammar problems and add more reference about the research region.
- We redescribe the data parts;
- We add the table with basic social-economical information of cities and regions to make the article more reliable;
- We add analysis of the weights in the results.
Again, we appreciate your time reviewing our paper and your comments which have been valuable to us. We would love to hear from you should there be any further questions or advice.
Best regards,
Authors
Reviewer 3 Report
Dear editor and authors,
I consider that this paper could be an interesting ms for this journal but there are several issues. Please, see my attached comments. Now, the English language is something weak. There are a lot of paragraphs without references. I won´t suggest any reference but you are encouraged to fill these lacks. A lot of parts do not have sources, this is key to be sure where the data come from. The elaboration of the index is ok, but where are the sources and references? The figures can be improved, see my comments. The discussion does not include any comparison to other studies with similar indexes, dynamics in other countries, regions as it was mentioned in the intro with India, for example. Please, see more comments in my attached pdf.

Author Response
Dear Reviewer:
We appreciate your detailed comments, which have greatly helped us polish our paper. We feel sorry for the defects in the previous draft submitted. We have modified our paper accordingly, as is shown in the paper with red text.
We answer your concerns as follows:
- I consider that this paper could be an interesting ms for this journal but there are several issues. Please, see my attached comments. Now, the English language is something weak. There are a lot of paragraphs without references. I won´t suggest any reference but you are encouraged to fill these lacks. A lot of parts do not have sources, this is key to be sure where the data come from. The elaboration of the index is ok, but where are the sources and references? The figures can be improved, see my comments. The discussion does not include any comparison to other studies with similar indexes, dynamics in other countries, regions as it was mentioned in the intro with India,
We correct all grammar problems that we can find and rewrite the abstract with the points you mentioned. We merge the introduction and literature review together and delete some irrelevant parts. We also adjust the rest parts with your suggestion. We add more reference in research regions, data sources and indicators construction. The information of local economy would be found in the new table. We also redescribe the data parts. All figures, numbers and formula has been reset to correct mode. The figure 1 and 2 have been merged together to increase readability. We rebuild the structure of discussion and add the compares with other regions. We also discuss some existing methods in discussion.
Based on other reviewers’ advice, we have modified the paper as follows:
- We change our title to“The Coupling and Coordinated Development from Urban Land Using Benefits and Urbanization Level: case study from Fujian Province (China)” as other reviewers suggested;
- The introduce of coupling degree and coupling coordinated degree would be found in introduction L70-78 to help readers comprehend these concepts;
- The sum of the weights showed in table 1 (original table 2) is equal to 1, the supplementary description would be found in L260-261;
- We add analysis of the weights and make some adjustments of the results.
Again, we appreciate your time reviewing our paper and your comments which have been valuable to us. We would love to hear from you should there be any further questions or advice.
Best regards,
Authors
Reviewer 4 Report
The authors begin by stating the problem: increasing demand for residential land due to population growth (urban expansion). This issue has been around for a very long time as the world population has doubled in less than 50 years. The authors then point the focus to Fujian province in China. They point out the natural barriers to growth there, such as mountainous terrain. At the end of the Introduction, the authors outline the rest of the manuscript by section.
Section 2 is a concise literature review.
In section 3 the authors turn to data and methodology. The authors included maps of the region in the Introduction and in section 3. They also produce equations that involve indexes. In section 3.3 they go into the components of the indexes and then in section 3.4 they go into index weighting. Sections 3.5 and 3.6 feature two models. I want to point out that the authors had referred to CCD in the introduction (line 67) without spelling out what these initials stood for. This is unacceptable and must be corrected. In line 308 the authors finally spell out what it means.
Section 4 includes the results, which the authors present in tables, figures and maps. There is quite a lot of information here, and the authors then move to a discussion in section 5.
The authors immediately begin the discussion by explaining the implications of the results toward urban expansion in the Fujian province. In the second line of the Discussion (line 403) we see what is a common problem in the manuscript: an English grammar mistake. This one is a subject verb agreement problem. It should say urban land use benefits are lagging instead of urban land use benefits is lagging. There are grammar mistakes like this throughout the manuscript. They must all be corrected before the manuscript can be published. The third sentence in the section is also incorrect grammatically. The points the authors are making are sound, but they get lost on the reader because of poor grammar. Also, the authors use the metaphor “on the one hand” way too many times. This makes for very confused reading. You must re-write the material from 406 on down and get rid of all the references to “on the one hand and on the other hand." The authors also need to break up this HUGE first paragraph into separate paragraphs to make the discussion easier to follow AFTER the English has been corrected. Start a new paragraph with the material at line 410. Then start a new paragraph with the material on line 413. Then start another new paragraph with the material on line 420 “Economic development often comes….. " Another paragraph should start with the material on line 431 “We also found ……"
The Conclusions section is concise and includes three bulleted points that appear to be rather straightforward.
Decision: Major revisions and re-submission required.
Author Response
Dear Reviewer:
We appreciate your detailed comments, which have greatly helped us polish our paper. We feel sorry for the defects in the previous draft submitted. We have modified our paper accordingly, as is shown in the paper with red text.
We answer your concerns as follows:
- The authors begin by stating the problem: increasing demand for residential land due to population growth (urban expansion). This issue has been around for a very long time as the world population has doubled in less than 50 years. The authors then point the focus to Fujian province in China. They point out the natural barriers to growth there, such as mountainous terrain. At the end of the Introduction, the authors outline the rest of the manuscript by section.
We merge the introduction and literature review together and delete some parts that not directly related to the research. We also adjust the rest parts to integrate with the research question by your suggestion.
- In section 3 the authors turn to data and methodology. The authors included maps of the region in the Introduction and in section 3. They also produce equations that involve indexes. In section 3.3 they go into the components of the indexes and then in section 3.4 they go into index weighting. Sections 3.5 and 3.6 feature two models. I want to point out that the authors had referred to CCD in the introduction (line 67) without spelling out what these initials stood for. This is unacceptable and must be corrected. In line 308 the authors finally spell out what it means.
We add more reference about the research region and merge figure 1 and 2 together. We describe coupling coordination degree (CCD) in L70-78 of introduction. According to your opinion, we deeply agree that we should mention it as earlier as possible for readers. We also correct the unclear abbreviation of CCD.
- The authors immediately begin the discussion by explaining the implications of the results toward urban expansion in the Fujian province. In the second line of the Discussion (line 403) we see what is a common problem in the manuscript: an English grammar mistake. This one is a subject verb agreement problem. It should say urban land use benefits are lagging instead of urban land use benefits is lagging. There are grammar mistakes like this throughout the manuscript. They must all be corrected before the manuscript can be published. The third sentence in the section is also incorrect grammatically. The points the authors are making are sound, but they get lost on the reader because of poor grammar. Also, the authors use the metaphor “on the one hand” way too many times. This makes for very confused reading. You must re-write the material from 406 on down and get rid of all the references to “on the one hand and on the other hand." The authors also need to break up this HUGE first paragraph into separate paragraphs to make the discussion easier to follow AFTER the English has been corrected. Start a new paragraph with the material at line 410. Then start a new paragraph with the material on line 413. Then start another new paragraph with the material on line 420 “Economic development often comes….. " Another paragraph should start with the material on line 431 “We also found ……"
We rewrite the discussion parts and improve the structure to increase readability. We also modified full article grammar. The discuss of the method of this research could be found in the discussion parts.
Based on other reviewers’ advice, we have modified the paper as follows:
- We change our title to“The Coupling and Coordinated Development from Urban Land Using Benefits and Urbanization Level: case study from Fujian Province (China)” as other reviewers suggestedï¼›
- We rewrite the abstractï¼›
- We add more reference about the research region and a new table about local economy information;
- We redescribe the data parts and explain the reason of the year we selected as a sampleï¼›
- The sum of the weights showed in table 1 (original table 2) is equal to 1, the supplementary description would be found in L260-261;
- We format the formulas and characters to increase the readability;
- We add analysis of the weights and make some adjustments of the results;
- We remake the figures to add more information.
Again, we appreciate your time reviewing our paper and your comments which have been valuable to us. We would love to hear from you should there be any further questions or advice.
Best regards,
Authors
Round 2
Reviewer 2 Report
Thank you for considering all my suggestions. Especially, I am glad about the increasing level of the maps.
Reviewer 3 Report
I consider that the authors improved a lot the paper. My unique concern is related to the figures. The graphs with bars: please, eliminate the lines inside the graphs, ticks inside, no borders and borders for the bars.
Reviewer 4 Report
It looks like the authors were thorough and diligent in making their revisions.